



# The weather today rocks or sucks for my tree: Exploring the understanding of climate impacts on forests at high school level through tweets

Thomas Mölg[1*], Jan. C. Schubert[2*], Annette Debel[1], Steffen Höhnle[2], Kathy Steppe[3], Sibille Wehrmann[1], Achim Bräuning[1]

*These authors contributed equally to this work*

[1]Institute of Geography, Friedrich-Alexander-University Erlangen-Nürnberg (FAU), 91058 Erlangen, Germany

[2]Lehrstuhl für Didaktik der Geographie, Friedrich-Alexander-University Erlangen-Nürnberg (FAU), 90478 Nürnberg, Germany

[3]Laboratory of Plant Ecology, Faculty of Bioscience Engineering, Ghent University, 9000 Ghent, Belgium

*Correspondence to*: thomas.moelg@fau.de or jan.christoph.schubert@fau.de

**Abstract.** With the progression of global warming, impacts on the human sphere will undoubtedly increase. One prominent example at mid latitudes is the stress of forests under climate change, which the project "BayTreeNet" (https://baytreenet.de/) addresses from an interdisciplinary viewpoint. Scientists from physical climatology, dendroecology, and educational research collaborate to examine how long-term changes in weather patterns affect the state of trees, and how the atmosphere/tree relation can be used to the advantage of improving the communication of climate change effects to, in particular, high school students. This article presents a 1-week case study from summer 2021, when a distinct variability in weather patterns induced significant tree responses. The students of seven selected partner schools commented these responses online through a framework including real-time weather and tree data as well as tweets, which was incorporated in their educational geography program. The results demonstrate that the students succeed in verbalizing the measured weather and, furthermore, manage to draw linkages to the stem diameter changes of trees. Problems arise with the use of less perceivable variables like the sap flow in the trees; also, the student posts exhibit shortcomings in establishing causal connections. Hence, the case study points to a discrepancy between describing basic environmental information and appreciating, or understanding, the underlying mechanistic links. This point will serve to refine future classroom concepts and, moreover, to enhance the communication of climate change effects on forests for the general public.

## 1 Introduction and general motivation

With the arrival of evening, TV prime time approaches and one show with it – the daily weather forecast. "The world's most watched science show" is, therefore, a statement that has appeared in the public and in the media. The implied importance of weather news has even increased with the rise of new, internet-based technologies over the past 20 years, which have led to a multiplication of weather information services; a related scientific study for the U.S. concluded, unsurprisingly, that "weather forecasts are a daily part of the lives of the vast majority of the U.S. public" (Lazo et al., 2009, p. 775). The roots of this circumstance go far back in time – let's think of Aristoteles' treatise "Meteorologica" or the century-long association of weather with the divine, which reflect the general desire of humans to understand weather. The side effect of this deeply-anchored interest is a familiarity of the public with



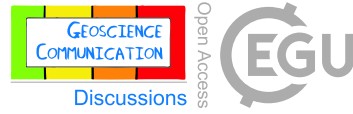

some central (scientific) terms like *high pressure*, *front*, and *cyclone*, and a sense of what they mean in practice
for local weather phenomena, or generally speaking for impacts (e.g., on one's own daily life, on health, or on the
environment). The scientific synthesis of these terms is expressed in the systematic study of large-scale daily
patterns in the atmosphere, which was mostly developed for central Europe in the 1940s/50s, and has to become
known by the German term "Großwetterlage" (see, for example, James (2007) for a short historical overview).
Although the scientific concept as well as the daily consumption of the topic in public media refer to *weather*,
meaning the state of the atmosphere ranging from a short moment in time to a few days, there is also a *climate*
dimension; consider that certain weather patterns become less or more frequent over decades and longer. The
consequences will be a change in the seasonal or annual characteristics of air flow patterns (which we call the
large-scale *atmospheric circulation*), and hence a change of the surface climate at locations in the reach of the
weather patterns. This relation is the starting point for connecting weather patterns to our changing climate, which
has been studied extensively in recent research (see Sect. 2). Importantly, the scientific weather pattern term mostly
encapsulates the distribution of air pressure centres and the resultant large-scale circulation modes, which are also
embodied by the elements that appear on weather maps in the daily news or on our cell phones as "H" or "L" and
wind arrows, feeding the public with the basics of an important climate change aspect. The present article
originates from a transdisciplinary project that aims to exploit the omnipresence of weather information in today's
world, and the increasing public awareness of climate change impacts on the environment, ecosystems, and human
health.
In this context, a prominent impact case is the response of forests to climate change. Trees are important organisms
for forest ecosystems and society as they provide ecological services, and tree-ring research (dendrochronology)
is a well-established discipline to assess the impacts of climate on tree growth and to reconstruct climate conditions
of the past. Although a number of concepts and tools have been developed for training the basics of tree-ring
research in different formats to the public, such as articles or videos (e.g., Davi et al., 2022), there is hardly any
tool to monitor the effects of weather on tree performance in real time, and to get the relevant explanations of the
climatic background. Here, we target this gap by making real-time weather and tree data available to high schools
in their geography programs, in order to investigate characteristics of climate change communication and education
at the high school level. While this provides the general motivation, Sect. 2 will explain the specific circumstances
as well as the structure of the research and the article.

## 2 Specific context and goals

The transdisciplinary project mentioned is called *BayTreeNet* and was described in a recent publication from a
conceptual viewpoint (Bräuning et al., 2022). The project was established in response to the evidence of increasing
drought stress on forests in Bavaria in the most recent decades (Debel et al., 2021) and, hence, presents an effort
to generate more knowledge of this threat with regard to future climates. *BayTreeNet* tackles the topic from the
perspectives of three special fields. Physical climatology, dendroecology, and educational research collaborate
under the overarching goal of increasing public awareness and preparedness for stress on forest ecosystems under
climate change, which must build on a solid communication and education of the problem. Besides the
transdisciplinary basis, a novel facet of *BayTreeNet* is to look at the climate/forest interaction at the scale of



weather patterns (henceforth GWL, according to the German term "Großwetterlage" introduced in Sect. 1), which
adds value to the traditional approach of relating forest responses to more aggregated climate data (e.g., monthly
or annual mean values). The GWL resolution, in turn, provides the connecting link to the public weather
information (Sect. 1), and is therefore a promising starting point for enhancing dissemination and education in the
given context.

In the physical climatology sub-project, GWL in the past and in the future, and their manifestation at the local
level, are examined with the help of meteorological observations and climate modelling. An important initial result
was the generation of a new climate data set for Bavaria (1987-2018) with a very high resolution in space (Collier
and Mölg, 2020). The dendroecology sub-project relies on a network of measurement sites over Bavaria (Sect.
3.1), where important tree growth variables are monitored. A first study using these data concluded that the tree
growth/climate relationships show an elevation dependency and, moreover, have not been constant over the past
five decades (Debel et al., 2021). The core of the educational sub-project, in turn, is a collaboration with high
schools in the vicinity of the measurement sites. Their students (grades 10 and 11 and 15-17 years old) observe
the weather and tree growth data in real-time, which is enabled by an accessible interface on the project webpage
(https://baytreenet.de/), and subsequently translate their interpretation of the trees' physiological responses to
simple Twitter (now known as X) messages. This translation embodies the educational aspect and provides the
data for the communication research of the present study.

If the synthesizing character is neglected, the three mentioned scientific fields can build on a rich literature with
regard to the present topic. Investigations in climatology have, for instance, targeted the role of GWL for health,
extreme events, hydrological and ecological processes (Bissolli, 2001; Post, 2002; Clark and Brown, 2013; Loikith
et al., 2017; Pisistaki et al., 2020; Zong et al. 2022). Given the concept's history (Sect. 1), central Europe has
always been a focus (e.g., Riediger and Gratzki, 2014; Herrera-Lormendez et al., 2021). In dendroecology, studies
of the dependence of tree growth on climatic variables in central Europe have an equally long history; some major
recent focus points were, for example, how summer temperature and rainfall variability affect the growth of
European tree species (Debel et al. 2021; Dulamsuren et al. 2017; Friedrichs et al. 2009; Kraus et al. 2016). And
finally, contributions from the field of educational research on climate change deal, among other things, with the
incorrect or insufficient ideas of students. For example, several conceptual models of the causes of the
anthropogenic greenhouse effect have been identified among students, which contradict scientific explanations
(including the wrong explanation of the greenhouse effect via the hole in the ozone layer) and can hinder learning
processes (Choi et al. 2010). With respect to consequences of climate change, it was shown that students can name
a variety of examples but have major problems with spatial differentiation (Boyes, Stanisstreet 1993; Leiserowitz
2021). Similar results were shown to concern many adults as well (Dove 1996).

However, to our knowledge no scientific project has brought together the three disciplines for the purpose of
fostering communication and education, regarding the important topic of forest response to climate change. In the
present article, we make a first effort to synthesize the multidisciplinary data through a 1-week case study for early
summer 2021. The main goal is to examine what level of understanding exists in high school students about the
forest/atmosphere interactions, by considering the associated Twitter data. Our working hypothesis is that students
are basically successful at verbalizing weather data, yet more difficulties exist with verbalizing tree data and, in





particular, with correctly describing the relationships between weather and tree data. This hypothesis is derived
from the considerations that weather reports and conversations about weather are a part of the students' everyday
life (cf. Sect. 1), while the terms related to the state of trees are much less known, so students have less practice in
verbalizing information on tree growth. Hence, linking weather and tree data represents the greatest challenge and
involves a comparatively high complexity.

To realize our approach, we selected a week with obvious variability in weather patterns and tree responses, and
at the same time, with a relatively high number of student tweets. Since identifying such a week involved screening
the physical-science data as well, we first present the climatological and dendroecological input from our project
in Sect. 3; this is kind of a preparatory step that is needed to comprehend the entire study. The actual
communication (educational) research is then presented in Sect. 4, before the conclusions in Sect. 5.
**3 Physical science basis**
**3.1 Data and methods**
**Weather pattern classification.** In climatology, daily values of sea-level pressure and geopotential height at 500
hPa (roughly, the mid-troposphere level) are the typical input variables for the classification of GWL. The most
common and popular classification was introduced and published by Hess and Brezowsky in 1952 in the Catalogue
of GWL in Europe (see James, 2007), containing daily weather patterns since 1881. The national German
Meteorological Service (DWD) has been using this classification since then. The manual classification of Hess
and Brezowsky yields 29 types of GWL, which are characterized by the air pressure pattern, cyclonicity, the main
direction of large-scale air flows, and vorticity (Werner and Gerstengarbe 2010). The distribution, intensity, and
characteristics of high- and low-pressure systems define the different types of GWL, which need to last at least
three consecutive days. GWL types show different occurrence probabilities during the year and vary among the
seasons, and each GWL can bring significantly different weather to certain regions and locations in Europe. Table
1 summarizes some relevant aspects of the traditional GWL scheme, which is the basis for the present study.

**Table 1:** Some examples of the 29 GWL (frequent ones and relevant ones for our study) in original German and
translated into English definitions after James (2007) and their mean annual probabilities (1881-2008) after Werner
and Gerstengarbe (2010).

| GWL | Original definition (German) | Translated definition (English) | Probability |
|---|---|---|---|
| WZ | Westlage, zyklonal | Cyclonic Westerly | 15.70 % |
| HM | Hoch Mitteleuropa | High over Central Europe | 8.89 % |
| BM | Hochdruckbrücke (Rücken) Mitteleuropa | Zonal Ridge across Central Europe | 7.72 % |
| NEa | Nordostlage, antizyklonal | Anticyclonic North-Easterly | 2.18 % |
| HNFa | Hoch Nordmeer-Fennoskandien, antizyklonal | High Scandinavia-Iceland, Ridge C. Europe | 1.14 % |


**Wood formation detection.** In the dendroecology sub-project, we have established a dendroecological network
of eleven study sites (initially ten) to monitor responses of forest ecosystems to changing climate dynamics at



different temporal resolutions. While long-term growth changes, adaptations of the wood anatomical structure and
adaptation of intrinsic water use efficiency are assessed at decadal to annual time scales, cambial growth dynamics
and stem diameter variations are monitored at interannual to even hourly resolution. While these dendroecological
sample sites are located inside forests, an additional monitoring method detects tree responses to changes in the
environment in real-time. For this purpose, single tree individuals close to each sample site were equipped with an
internet-based logger and sensors to measure sap flow and stem diameter variations. The sap flow sensor measures
the sap flow transport rates ($cm^3 hour^{-1}$) from the roots to the crown (Smith and Allen, 1996; Steppe et al., 2010,
2015; Vandegehuchte and Steppe, 2013). The point dendrometer registers stem diameter changes (mm) caused by
the reversible shrinking and swelling of living stem cells in 20 minute intervals. These are visible as daily cyclic
stem diameter variations, the amplitudes of which inform us about the water status of the tree (Deslauriers et al.
2007; Drew and Downes 2009; Steppe et al., 2015). In addition, the dendrometer records the irreversible growth
of the xylem and phloem by the formation of new xylem (wood) and phloem (bark) cells, which becomes visible
as a long-term trend of increasing stem diameter superimposed on the daily radial variations. While under humid
conditions, the long-term trend is positive, tree stems may shrink in diameter to a certain extent under drought
conditions.

**Data preparation.** These two high-precision instruments indicate the individual trees' responses to varying GWL
and the associated weather conditions like, for example, a heavy rain event or a heatwave. The measured data are
transmitted real-time to the PhytoSense Cloud service, which is a tool for data storage, analysis, processing, and
running model simulations to visualize the trees' hydraulic function and carbon status (Steppe et al., 2016). Finally,
the processed data are sent to the project homepage (https://baytreenet.de/), where the physiological data of each
"talking tree" are graphically displayed. At the same time, information about the current weather at each site, e.g.,
precipitation and temperature, is available on the homepage. This includes a map of central Europe with the current
flow patterns at the 500hPa (~5km) and 850hPa (1.5km) geopotential height levels, and a weather map indicating
the location of fronts. Therefore, the short-term responses of trees to weather and long-time tree growth become
detectable and displayable.
**3.2 Results: Weather variability and tree response**
The calendar week 22 in early summer 2021 (May 31st to June 6th) was detected appropriate for this case study.
In this week, two GWL were present: NEa (Antycyclonic North-Easterly) on Monday, Saturday, and Sunday (May
31st, June 5th and June 6th) and HNFa (High Scandinavia-Iceland, Ridge Central Europe) from Tuesday to Friday
(June 1st to June 4th). Both GWL are typical weather types for the early summer season and show their maximum
likelihood in May (HNFa) and June (NEa). Measured against the full year they occur with a probability of 1.41 %
(NEa) and 2.18 % (NEa) (cf. Table 1). The two GWL belong to anticyclonic types and typically induced the highest
average air temperatures in the recent past. For example, HNFa shows a deviation of +4.36 °C in the daily mean
air temperature with reference to the period 1951-1978 (Werner and Gerstengarbe 2010). Considering
precipitation, both GWL coincide with drier conditions on average, however, the drying signal is not as significant
as the warming signal.

These average anomalies are consistent with the air temperature characteristics in June 2021, as seen in the
measurements at the official DWD weather stations. In this month, particularly high temperatures were recorded

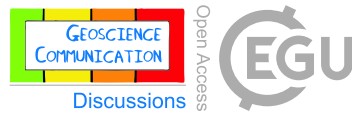



throughout Germany and daily anomalies of up to +4.5°C (compared to the 1961-1990 mean) occurred locally.
For Southern Germany the situation was very similar. In the case study week, the air temperature anomaly
increased over time until June 3rd and June 4th, when it reached the maximum of +4.9°C departure (same reference
period), and subsequently decreased until the end of the week. These characteristics are also reflected in four
stations close to our tree sites (Fig. 1), which were selected on the basis of reliable tree data and sufficient Twitter
messages. Precipitation at the 35 DWD weather stations in Southern Germany shows a steep increase from June
3rd with its maximum on June 5th and a subsequent decrease on June 6th. At three of the four sites investigated,
a precipitation event is recorded on June 5th with values up to 20 mm (Wunsiedel), while at the fourth site
(Veitshöchheim) this occurs one day later with more than 30 mm (Figure 1).

To summarize the case study week, strong positive air temperature anomalies were measured in Southern Germany
and Bavaria (after cold phases in May), reaching their maximum roughly in the middle of the week. Shortly after
this temperature maximum, precipitation occurred after several days of drought. Thus, the selected 1-week window
was characterized by a transition to warmer and wetter conditions.

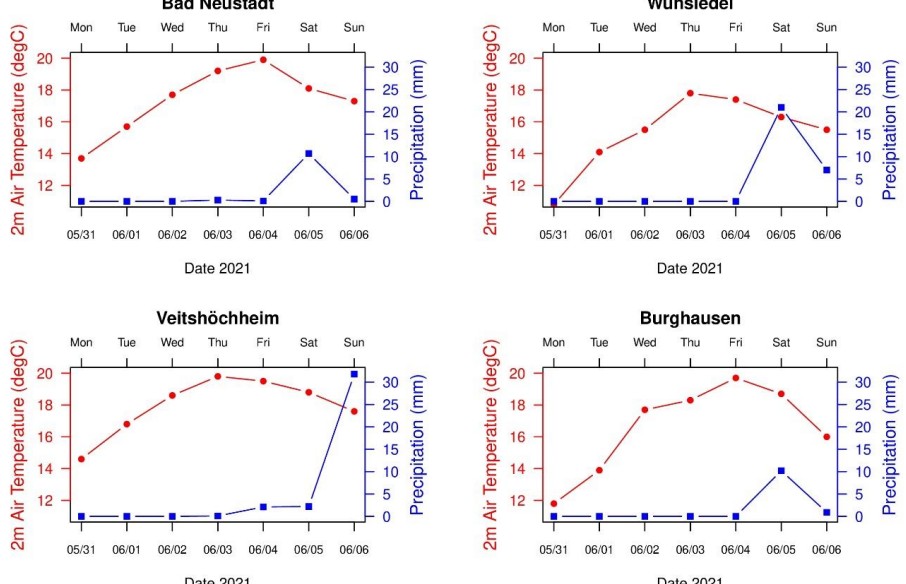


**Figure 1:** Two-metre air temperature (dot symbols) and precipitation (rectangles) at four selected weather stations
in Bavaria during the calendar week 22 of the year 2021 (data basis: DWD).

How did these atmospheric conditions affect the sample trees? Generally speaking, most talking trees showed
similar responses to the prevalent weather. The tree responses only differed in the timing between different
locations, since the rain events happened asynchronously between sites, as discussed above.

From June 1st to 4th, the influence of the HNFa weather pattern (with its dry and warm weather conditions) led to
characteristic tree physiological response patterns (e.g., Steppe et al., 2015) that indicate sufficient water supply

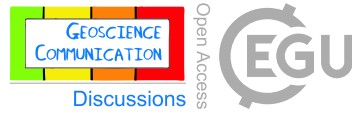

without water shortage (Fig. 2). Initiated by transpiration and regulated by the stomata, the water transport from
the roots to the crown started shortly after sunrise. Consequently, the sap flow rate increased rapidly in the early
morning, peaked around noon, and remained at high level until the evening. After sunset, the sap flow rate dropped
to pre-dawn level, showing minimum values in the early morning hours before the beginning of the subsequent
day's cycle.

The dendrometers recorded daily cycles of diameter changes, mainly due to the reversible shrinking and swelling
of the stem caused by the depletion and replenishment of the stem water stored in wood and bark tissues. Stem
diameters decreased in the morning shortly after sunrise, since photosynthesis started and water transpired through
the open stomata. As long as transpiration persisted and water flowed from the roots to the crown, the water
reserves in the stem depleted and the stem diameter shrank. After sunset, stem diameter began to increase again
due to the cessation of transpiration and the refilling of water reserves in the stem (Steppe et al., 2006). Beside
these daily cycles, the maximum stem diameters increased from day to day because of irreversible stem growth
driven by turgor and formation of new xylem cells.

With the occurrence of rather strong rain events later in the week, both data series show a significant change (Fig.
2). On the one hand, the sap flow rate of most (7 out of 10) trees did not increase to the same degree as in the days
before, and furthermore, the sap flow of all talking trees decreased earlier around midday. A high cloud cover
fraction and high air humidity reduced transpiration, hence the stem water transport declined during and after the
rain event. On the other hand, the talking trees overwhelmingly showed a sharp increase in diameter shortly after
the rainfall. Moreover, the diameter curve showed no decline during the daytime, pointing to continuous refilling
of the internal water storage pools that were emptied during the preceding dry period. Thus, higher air humidity
and less vertical water transport caused intensified water storage in stem cells. As a result, the cells got water-
saturated, and the depletion effect (causing a decrease in diameter) was mitigated.

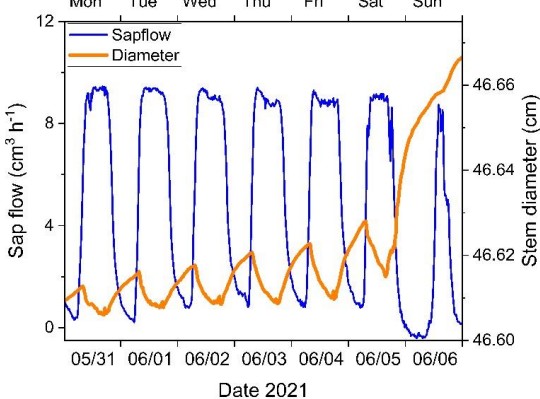


**Figure 2:** Stem diameter variations and sap flow rates of *Tilia platyphyllos* at Burghausen as an example of tree
response during the period May 31st to June 6th, 2021. Note that from June 5th to 6th, precipitation occurred,
leading to a strong increase in stem diameter.





## 4 Communication and educational research

### 4.1 Methodology

The participating students took part in an online introductory course, which served to build a basic understanding of GWL as well as of tree growth conditions. In addition, exercises were included on how to interpret tree data, in particular, what does (high/low) sap flow mean and what is meant by the variation in stem diameter of a tree? Together with the available information on current weather patterns, the students at each participating school were asked to translate the information on the tree's state as well as the connections with the weather situation into linguistic messages in the form of tweets: short, concise, in everyday language.

For this case study, we analysed the students' tweets during the carefully selected week in early summer 2021 (Sect. 3) with one guiding question: To what extent are the tweets (as linguistic translations of tree data and weather data) appropriate to the subject? The methodological procedure for evaluating the tweets involved several steps. As part of the preparation for the data analysis, the tree data as well as the tweets for all sites were reviewed, and sites where central data were missing were excluded. For the investigated period, one site was excluded due to missing tree data (Bad Reichenhall, sensor error) and another site where no tweets were made (Ettal). Another site (Immenstadt) could not be considered because tweets were only available for three days and, in addition, the tree data provided only limited meaningful information as the tree on site was dying.

For the remaining seven sites, the weather data and the tree data were prepared in the form of diagrams (e.g., Sect. 3) and the students tweets were converted into an editable version suitable for coding. Emojis that did not influence the statement but were merely illustrative in nature were not considered in this process. Several evaluation or analysis steps followed, oriented along the principles of a "thematic analysis" (Braun and Clarke, 2006):

(1) Becoming familiar with the data: Through repeated and close reading of the tweets, also against the background of weather data and tree data (as described in Sect. 3.2), an overview and a deeper understanding of the tweets were created by the researchers. This also involved description of the frequencies of tweets per site and per day.

(2) Coding with initial codes: The data material was coded according to five deductive categories. These included statements about (i) weather, (ii) growth/stem diameter, and (iii) sap flow. In addition, statements in which students (iv) combined two or more of these categories (for example, effects of weather on tree growth/stem diameter) were coded in a separate category. The same was true for (v) statements in which there was an exchange between students from different tree sites.

(3) Search for patterns and salient features in the categorized data: Within the categories, statements were screened for characteristic commonalities or patterns. In addition, conspicuous features were examined. The guiding principle here was a comparison with the scientific basis, i.e. the weather data and tree data in the form available to the students on our homepage. This interpretative step was initially undertaken independently by two researchers, who then compared the interpretations and attempted to reach a consensual interpretation in the event of discrepancies. The results of this step were, in turn, presented and discussed in the working group, which is linked to the goal of reducing subjectivity that is present in interpretative steps.

(4) The results of the analyses and interpretation were summarized, written down and supplemented in many places with illustrative quotes from the tweets (see Sect. 4.2).


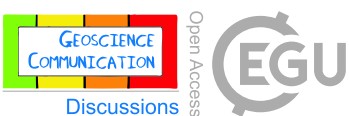

**4.2 Results: Interpretations by high school students**

The initial analysis of the tweets showed that three of the seven schools had at least one tweet for each of the seven days of the study period. For the four other schools, tweets were not available for every day, but for at least five days of the study period. In total, the data corpus included 43 tweets (Tab. 2). All 43 of the tweets contained statements about growth or stem diameter changes, 39 contained statements about sap flow, and 37 contained statements about weather. In 13 tweets, connections were made between at least two of the three elements. A sample tweet is shown in Fig. 3.

**Table 2:** Overview of tree locations and tweets.

| Locations of the "Talking Tree" | Number of tweets | Number of days with tweets |
|---|---|---|
| (1) Neuschönau | 7 | 7 |
| (2) Bad Neustadt | 7 | 6 |
| (3) Veitshöchheim | 7 | 7 |
| (4) Wunsiedel | 7 | 7 |
| (5) Tennenlohe | 5 | 5 |
| (6) Burglengenfeld | 5 | 5 |
| (7) Burghausen | 5 | 5 |

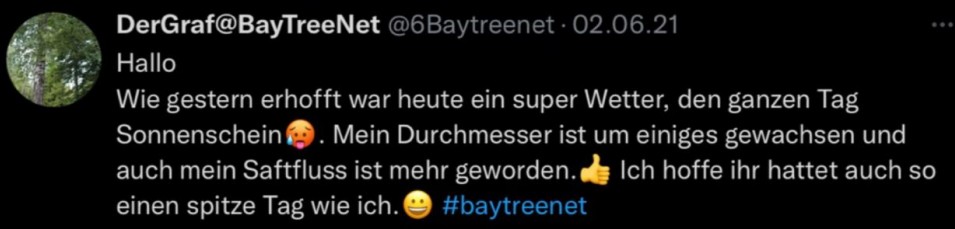

**Figure 3**: Sample tweet for illustrative purposes at the location Neuschönau, meaning "Hello. As hoped for yesterday, today was great weather, sunshine all day. My diameter has grown by quite a bit and my sap flow has also increased. I hope you had a great day like me." (BayTreeNet Neuschönau, 2021).

The following conspicuous features and characteristics were identified during the more in-depth analyses of the tweets.

**(1) Statements about the weather.** The statements about the weather are almost universally in line with the weather data; even the precipitation events that occurred after previous sunshine are correctly recognized and named ("Today was another sunny day" [Heute war wieder ein sonniger Tag] // "In any case, it rained and thundered for all it was worth at my place" [Bei mir hat es jedenfalls geregnet und gewittert, was das Zeug hält], Wunsiedel). In general, with regard to the weather, it can be seen that not all weather elements that are recognizable for the students are consistently addressed in the tweets and, moreover, relevant for the tree responses. Regarding



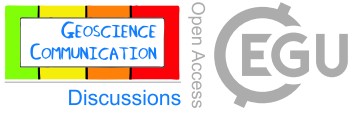

basic weather elements, references to sunshine and cloud cover, to dryness and precipitation, and to cold and warmth would have been possible. In the tweets, however, predominantly only one of these elements was addressed, only in individual cases at least two weather elements were cited together ("Since it was pleasantly warm again today, I was in a good mood despite Monday, even though it rained today and it was quite cloudy.", Tennenlohe). In addition, at some locations, different weather elements were mentioned when comparing the days. Thus, on some days there were comments (only) about the temperature, on other days (only) about the sunshine.

**(2) Statements on stem diameter.** The growth of the trees or changes in the stem diameter were consistently taken up in all tweets. The central common feature of most tweets is that a statement was made about the (thickness) growth of the tree stem or, more generally, that the tree had grown. However, the recognition or interpretation of the diurnal fluctuations of the stem diameter seems to be problematic. With very few exceptions ("Well, my diameter always goes steeply downhill during the course of the day, but I've grown a bit anyway" [Also so im Laufe des Tages geht es ja mit meinem Durchmesser immer steil bergab, aber ich bin trotzdem ein bisschen gewachsen], Veitshöchheim), these fluctuations were not discussed at all; instead, a single value of the stem diameter was often compared with a single value from the previous day. In some cases, there were incorrect interpretations from a scientific point of view, because it was not the minimum and maximum stem diameter of one day that were compared with those of the previous or next day, but individual values at different times of the day.

**(3) Statements on sap flow.** There are a similar number of tweets on sap flow as on weather. It is noticeable that the sap flow was more often described in a relatively vague and superficial way compared to the weather and the stem diameter. A typical example of what was frequently found is "The sap flow is good" [Der Sapflow ist gut]. With very few exceptions ("My sap flow was greater today at noon than yesterday, only dropping over the afternoon" [Mein Saftfluss war heute Mittag größer als gestern, nur über den Nachmittag hin ist er gefallen]; Neuschönau), diurnal variations in sap flow were not picked up in the tweets. Instead, individual values or maximum values of different days were compared (similar to stem diameter).

**(4) Links between weather, stem diameter and sap flow.** Linkages in the sense of causal relationships between weather, growth/stem diameter, and sap flow were rarely the subject of tweets. In most cases, the information on the three areas was rather unrelated to each other ("Hey friends. Today was finally awesome weather. My sap flow is shooting through the roof and I'm thinning out a bit. I hope you're all well" [Hey Freunde. Heute war ja mal endlich Bombenwetter. Mein Saftfluss schießt durch die Decke und ich werde ein bisschen dünner. Ich hoffe euch geht's gut], Burglengenfeld // "Hello. As hoped yesterday, today was great weather, sunshine all day. My diameter has grown by quite a bit and my sap flow has also increased" [Hallo. Wie gestern erhofft war heute ein super Wetter, den ganzen Tag Sonnenschein. Mein Durchmesser ist um einiges gewachsen und auch mein Saftfluss ist mehr geworden], Neuschönau). If links were made, the typical pattern was to start with the weather and go to the diameter or the sap flow ("But at least I was able to grow due to the rain" [Aber durch den Regen konnte ich wenigstens wachsen], Burglengenfeld // "My diameter has grown a gigantic amount due to the great weather" [Mein Durchmesser ist durch das tolle Wetter nochmal ein gigantisches Stück gewachsen], Tennenlohe). Links between sap flow and diameter were not made.

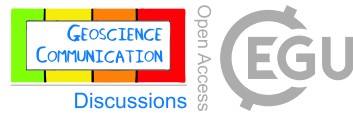

**(5) Communication between the tree sites.** In many cases, the other sites or the other trees were addressed in the salutation or the closing words ("Hello friends" [Servus Freunde], "Hello my dear fellow trees ["Hallöchen meine lieben Mitbäume]" as well as "Well, we'll hear each other tomorrow" [Naja wir hören uns dann morgen], Burghausen). In some cases, questions were also addressed to the other tree sites ("How was everything going with you?" [Wie war es denn bei euch?], Wunsiedel). In contrast, an explicit take-up of such questions as well as of content-related information on tree data and weather data shared by other tree sites occurred with very few exceptions (" Hi Karl. I think your wish for rain came true." [Hi Karl. Ich glaube, dein Wunsch nach Regen ist in Erfüllung gegangen], Wunsiedel). An exchange on measured values on the level of tree data was not detectable at any site.

In terms of summary and discussion, it can be stated that the students implemented the content of the online introductory course appropriately and well, at least on the formal or surface level. The number of tweets was relatively high (= daily) for most schools during the analysed period, and the tweets commented relatively consistently on weather, stem diameter/tree growth, and sap flow (as the three key metrics). The change in weather on June 5th was recognized and commented on. Thus, the central goal of having students verbalize measured data was achieved. With a detailed look at the tweets as well as in comparison with the tree data and weather data available for the respective location and period, it became clear that the tweets were not comprehensive (weather) or only considered parts of the measurement data (growth, sap flow). In particular, the fluctuations in stem diameter and sap flow over the course of the day, considered individually, seem to be too complex for the students to be addressed in their tweets. The connections between sap flow and stem diameter, which are even more complex, were not addressed by the students at any point. In contrast, weather elements were used (although only in a few instances) to show effects on the trees. This approach is scientifically correct and in line with the students' perceptions: Weather and weather elements are known to students from their own experience and from weather reports, see Sect. 1, and they can also feel them with their own senses. In contrast, changes in stem diameter are not perceptible with their own senses, at least within the resolution that occurred in the case study period. The same applies to the sap flow of trees. Both are only accessible via the sensors and graphical illustrations of the measured values. Starting from the weather, which is familiar and perceptible from everyday life, the students have tried to explain the tree data.

The goal of initiating a communicative exchange between students from different tree locations was only partially successful. Although there was almost always an address to other locations, the exchange was almost without exception on a superficial level. However, there was almost no exchange or discussion of weather data or measurement data between the sites. One reason could be that even understanding one's own tree data is challenging and not always successful. Thinking through data from other trees would mean an additional increase in demand and effort.

The results presented are subject to a number of limitations. Firstly, we only looked at a limited time period, which is not necessarily characteristic of other time periods. In addition, there were technical problems (sensor failure at one site) and not all schools sent out tweets during the week studied. The medium Twitter and the associated restriction to a maximum of 140 characters per tweet played a special role. On the one hand, this allowed the measured values to be verbalized by the students in a condensed manner and an accessible one for the interested

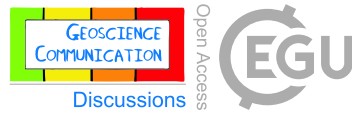

public. At the same time, discussing profound causal links between tree and weather data as well as a
communicative exchange on the measurements between the sites are made more difficult by this format alone. In
retrospect, it would have been useful to explicitly encourage students to submit multiple tweets within one day. At
the same time, from a realistic perspective, it must be stated that looking at the tree data and tweeting already
meant an additional task for the students. This made it difficult to increase the requirements further. Overall,
therefore, the fact that links between weather, stem diameter, and sap flow, as well as between tree locations, occur
very infrequently cannot be clearly or solely attributed to comprehension problems or the like; one reason may
also be the format (140 characters). Although in some places it can be assumed that there were indeed difficulties
in comprehension, it can be seen as positive that the students succeeded in transferring the tree data into short
tweets in everyday language.
**5 Conclusions and Outlook**
If we close the circle to the introduction of this article (Sect. 1), the 1-week period in summer 2021 was a good
choice in the sense that it showed ample weather variability; warming in the first half was followed by strong rain
events at our study sites, which induced significant tree responses. These were characterized by a growth in stem
diameter and a high sap flow rate, while pronounced diurnal cycles were superimposed on these weekly tendencies.
In connection with the tweet data, a first idea of how high school students understand the relationship between
atmospheric variability and tree responses can be shaped.
The results demonstrate that students put a focus on weather elements, which underlines the connection of people
to daily weather information emphasized in the prologue. Regarding the tree variables, it seems that they pick up
the stem diameter changes more easily than sap flow variability. This is not uncommon, and may have several
reasons: compared to sap flow, stem diameter is easier to perceive, whereas sap flow is more elaborate and abstract.
In addition, trees with trunks of different thicknesses are part of our lifeworld context, presumably more prior
knowledge is available for stem changes than for sap flow. New information can be embedded in existing mental
models more easily as long as the new information is consistent with the existing mental model. Such an
importance of contexts and prior knowledge is well known in cognitive educational science (for example
Podschuweit and Bernholt, 2020; Witherby and Carpenter, 2022). One common problem irrespective of the tree
variable was the appreciation and understanding of the distinct diurnal cycles in stem diameter change and sap
flow. In the authors' experience from teaching at university level, understanding the meaning of such daily cycles
(as a form of internal climate variability) appears to be hard in general. Another common problem was that causal
relationships between the various variables were hardly made, except some basic characteristics of how weather
tendencies affect the tree responses. While this represents without doubt a more complex question, the deficiency
also suggests that teaching the climate as a physical cause/effect system could probably be enhanced at school
level. Altogether, the results confirm our working hypothesis and point to a discrepancy between verbalizing
environmental information and drawing mechanistic links.
In the near future, it is planned to complete the study of systematic GWL changes over many decades, which will
allow the students to relate the topic more strongly to the climate change aspect. We will also attempt to maintain,
and even to extend, the tree network in the future, either by a follow-up research project or by other means of

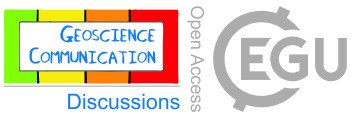

public education. For future classroom sessions, teaching materials that focus on the relationship between weather
conditions and the responses of trees are currently developed and evaluated. These connections are to be worked
out both in general and in concrete case studies. For the latter, exemplary periods with tree data and tweets of the
students will be included.

**Data availability.** Weather data are openly available from the German Weather Service (DWD). The tree data and
associated tweets can be obtained from our website at https://baytreenet.de/.
**Author contributions.** AB, TM and JCS designed the study and arranged the project funding. AD, JCS, SH and
SW analysed the data. KS supported the tree sensor data collection. All authors contributed to the writing and
editing of the manuscript, which was coordinated by TM.
**Competing interests.** The authors declare that they have no conflict of interest.
**Ethical statement.** Ethical approval for this research was given by the Friedrich-Alexander-University (FAU)
Erlangen-Nürnberg ethics committee.
**Acknowledgements.** This research was funded by the Bavarian State Ministry of Science and Arts, as part of the
Bavarian Climate Research Network (bayklif). We thank Katrien Schaepdryver, supported by the BOF research
project TreeWatch (grant no 01J07919), and Erik Moerman from Ghent University for their technical support. In
particular, we thank the teachers and students of the partner schools for their big engagement in this project.

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
