# Peer review of "understanding of climate impacts on forests at high school level"

_Geoscience Communication, 2023_

## Referee Comment (RC1)

Comments on GC-2023-5

In general, I like this paper and what the authors attempted to do. The idea of having real-time data available to the public and then working with them to interpret those data is a great idea. The paper is also well written. There are, however, three connected aspects of the paper that I think need to be clarified. The first is encapsulated in the very sentence that the authors present at the bottom of page 2, which is the idea of building on a "…solid communication and education of the problem". What is not clear in the description of the paper is how the authors communicated with the students and educated them. There is mention of a course on geography, but what is not clear is how the authors participated in that course or to what extent there was any preparation on the part of the authors with the students to understand what was being asked of them. The second aspect of the paper is the use of Twitter/X with high school students. The authors noted that the response rate was quite low. Is that because Twitter/X is not the main mode of communication among high school students? In other words, did the authors attempt to communicate with the students in the fashion that works best for the students (TikTok?)? The third aspect of the paper that needs clarification is why only one week of the project was chosen. This is the most problematic part of the paper. There is no sense of whether this week was at the beginning, middle or end of the geography course. Why only one week? Why not several weeks, maybe dispersed throughout the semester to see whether understanding increased or not? If the week chosen was at the beginning of the semester or the only time the authors communicated with the students, then it is not surprising that the students were superficial in what they communicated in their tweets. If tweeting is not their main mode of online communication, then it is also not surprising. Thus, it is not clear from the results whether the authors were measuring the effect of Twitter (i.e. is it the appropriate medium for online communication for teenagers), or the effect of timing, or the effect of understanding. To help with this issue, it would be helpful to understand the context of the week of data collection to the course it was linked to or to whatever program the authors had with the students to work through the connections between climate and tree responses. If the authors collected data on student understanding at different times, then that should be included in the paper. If they only collected data during one week, then we need to understand why only one week was chosen and what the context of that week was from the educational perspective (the climate perspective is adequately explained). We understand what was unique about the week presented from a tree response and weather perspective, but not in the context of the learning perspective. I think that if these points could be clarified, the paper would be strengthened and good to go for publication.

---

## Referee Comment (RC2)

Dear Editor;

I have read the manuscript entitled " The weather today rocks or sucks for my tree: Exploring the understanding of climate impacts on forests at high school level through tweets" and have provided some general comments below.

The manuscript topic is interesting and has merit and will attract many readers. There are numerous studies investigating the physiological response of forests to meteorological variables and impact of climate change on forests. However, since the results of these studies are mostly interpreted by researchers who are experts on the subject, they do not show how the public interprets the relationship between meteorological parameters and climate change and forests. For these reasons, the fact that the study targets the young generation, which will be heavily affected by the negative effects of climate change in the future and will play an important role in the mitigation against climate change, increases the importance of the study.

At the same time, I believe the manuscript needs more elaborated description, especially for the methods. In particular, the details of the education that high school students received within the scope of the study in their geography lessons and whether they were educated to understand especially tree diameter changes and sap flow cannot be understood from the text. On the other hand, it was stated that the students tweeted a small number of times, but the information about their motivation for tweeting was not included in the text. Are students completely free to tweet? Or were students told that their tweets would be used for a scientific study and therefore they were expected to tweet frequently?

Another point that limits the success of the study is that the selected period included only one week. It appears that this week was chosen because it involved a sharp variation in meteorological parameters and therefore a difference in the response of the trees. However, here the question of whether the selected week was at the end of the semester comes to mind. An examination to be carried out over a longer period would perhaps reveal that as the education the students received increased, they improved in their verbal expression of meteorological parameters and the response of trees to changes in these parameters.

Apart from the mentioned points, it would be appropriate to correct some minor errors, such as in the caption of Figure 1 (metre-by-meter). I also suggest mentioning the brand and the type of the devices used for physiological measurements in the relevant section. The resolution of the graphics in the preprint version I downloaded was quite low. I think it would be useful for the readers to check the resolution of the figures used in the text and increase it if possible.

---

## Author Response (AR1)

*Dear Editor,*

*Thank you for inviting us to submit the revised version of our manuscript. For the point-by-point reply below, we take the text from our final response as the basis (green) and indicate in yellow where changes in the revised MS were announced. In addition, we describe in the boxes below how we implemented them or what other changes we made. We hope this will allow you to comprehend the revision actions in a straight-forward way.*

*Thanks for your consideration.*

*Sincerely,*
*Thomas Mölg, Jan Christoph Schubert, & co-authors*

**Reviewer 1**

In general, I like this paper and what the authors attempted to do. The idea of having real-time data available to the public and then working with them to interpret those data is a great idea. The paper is also well written. There are, however, three connected aspects of the paper that I think need to be clarified. The first is encapsulated in the very sentence that the authors present at the bottom of page 2, which is the idea of building on a "…solid communication and education of the problem". What is not clear in the description of the paper is how the authors communicated with the students and educated them. There is mention of a course on geography, but what is not clear is how the authors participated in that course or to what extent there was any preparation on the part of the authors with the students to understand what was being asked of them.

RESPONSE-1: This is a valid point. Our team indeed briefed the students at each school before the start of the data commenting, which was described in the original manuscript at the beginning of Section 4.1. We propose to extend this description in the same paragraph and make the point of the initial training phase clearer.

CHANGES MADE: In Section 4.1, we inserted two sentences at the beginning of the first paragraph that explain the introductory workshop and its implementation in more detail. In addition, a new paragraph (second paragraph in Section 4.1) has been added to clarify the actual implementation and organization after the introductory workshop.

The second aspect of the paper is the use of Twitter/X with high school students. The authors noted that the response rate was quite low. Is that because Twitter/X is not the main mode of communication among high school students? In other words, did the authors attempt to communicate with the students in the fashion that works best for the students (TikTok?)?

RESPONSE-2: Twitter seemed particularly suitable for this project for several reasons. On the one hand, we wanted to accomplish the challenge of converting data into language, which fits well with Twitter as a primarily language-based medium. Secondly, the limited number of characters per message that Twitter had implemented at the time of the project required pointed and concise writing with a focus on

the central aspects. From a learning perspective, both together seemed very suitable for encouraging students to transform data into language. At the same time, Twitter enables simple, low-threshold communication with each other, which fitted in well with the idea of letting the trees communicate with each other. In addition, it seemed easier to attract the attention of a wider public via twitter than via conventional websites. Other social media such as TikTok would probably have been closer to the students' everyday lives, but the focus on visual or auditory content would have been less suitable for the project's objectives.

We would be happy to include more justification in the revised paper for the decision to use Twitter.

> **CHANGES MADE**: We added the reasons for the decision to work with Twitter (instead of a website or other social media) in the project as a new paragraph in Section 4.1 (third paragraph).

The third aspect of the paper that needs clarification is why only one week of the project was chosen. This is the most problematic part of the paper. There is no sense of whether this week was at the beginning, middle or end of the geography course. Why only one week? Why not several weeks, maybe dispersed throughout the semester to see whether understanding increased or not? If the week chosen was at the beginning of the semester or the only time the authors communicated with the students, then it is not surprising that the students were superficial in what they communicated in their tweets. If tweeting is not their main mode of online communication, then it is also not surprising. Thus, it is not clear from the results whether the authors were measuring the effect of Twitter (i.e. is it the appropriate medium for online communication for teenagers), or the effect of timing, or the effect of understanding. To help with this issue, it would be helpful to understand the context of the week of data collection to the course it was linked to or to whatever program the authors had with the students to work through the connections between climate and tree responses. If the authors collected data on student understanding at different times, then that should be included in the paper. If they only collected data during one week, then we need to understand why only one week was chosen and what the context of that week was from the educational perspective (the climate perspective is adequately explained). We understand what was unique about the week presented from a tree response and weather perspective, but not in the context of the learning perspective. I think that if these points could be clarified, the paper would be strengthened and good to go for publication.

RESPONSE-3: Thank you for the remark. The selection of the week was truly guided by the atmospheric conditions and the tree responses, with the aim of including a significant change in weather. We deliberately did not choose from the perspective of the educational project because the circumstances at the schools were very different. For example, some schools changed the pupils who wrote the tweets after short periods of around two weeks. At other schools, these changes only took place later. This was at the discretion of the schools. As a result, pupils with different levels of experience and motivation tweeted at the respective schools; at the same time, we cannot trace which pupils tweeted when. Therefore, we cannot investigate longer-term developments with regard to the quality of the tweets. Rather, it seemed sensible to us to select a limited period from an atmospheric-science perspective (due to a lack of information on the tweeting pupils). We propose to elaborate on these circumstances in more detail in the revision and take it up under "Limitations" (and consider making a new subsection to give the topic more weight).

Regarding data collection, there is now a huge amount of student data, presentation of which would be beyond the scope of this paper. These data will be analyzed in future studies; we will try to analyze several periods and to compare them. We ask for your understanding that we cannot include them here.

**CHANGES MADE**: On the one hand, we now include and explain in detail the different constellations of the participating schools/students (second paragraph in Section 4.1, see comment above). On the other hand, we also addressed this issue under "limitations" (4.3, first paragraph) and in "5 Conclusions and Outlook" (last sentence).

**Reviewer 2**

Dear Editor;

I have read the manuscript entitled "The weather today rocks or sucks for my tree: Exploring the understanding of climate impacts on forests at high school level through tweets" and have provided some general comments below.

The manuscript topic is interesting and has merit and will attract many readers. There are numerous studies investigating the physiological response of forests to meteorological variables and impact of climate change on forests. However, since the results of these studies are mostly interpreted by researchers who are experts on the subject, they do not show how the public interprets the relationship between meteorological parameters and climate change and forests. For these reasons, the fact that the study targets the young generation, which will be heavily affected by the negative effects of climate change in the future and will play an important role in the mitigation against climate change, increases the importance of the study.

At the same time, I believe the manuscript needs more elaborated description, especially for the methods. In particular, the details of the education that high school students received within the scope of the study in their geography lessons and whether they were educated to understand especially tree diameter changes and sap flow cannot be understood from the text. On the other hand, it was stated that the students tweeted a small number of times, but the information about their motivation for tweeting was not included in the text. Are students completely free to tweet? Or were students told that their tweets would be used for a scientific study and therefore they were expected to tweet frequently?

RESPONSE-4: This point is very similar to the issue raised by Reviewer 1, which strengthens our idea to describe the special, initial training at each school more extensively (see RESPONSE-1). At the same time, we will clarify the motivation. Basically, the students posted the tweets in their free time and the monitoring of the talking trees was not part of the formal lessons and grading. At the same time, the mentoring teachers asked the students to tweet once a day if possible.

**CHANGES MADE**: We implemented the comments on the workshop in Section 4.1 (first paragraph), where we explain this in more detail now. We also addressed the voluntary nature of participation in Section 4.1 (second paragraph) by explaining the circumstances at the schools.

Another point that limits the success of the study is that the selected period included only one week. It appears that this week was chosen because it involved a sharp variation in meteorological parameters and therefore a difference in the response of the trees. However, here the question of whether the selected week was at the end of the semester comes to mind. An examination to be carried out over a

longer period would perhaps reveal that as the education the students received increased, they improved in their verbal expression of meteorological parameters and the response of trees to changes in these parameters.

RESPONSE-5: We agree that the timing of the case study week with regard to the students' and high school schedule needs more discussion, see RESPONSE-3 above.
* * *
**CHANGES MADE**: The background of the case study week is explained in more detail in the newly added second paragraph in Section 4.1. In addition, possible limitations are now discussed in more detail in the new Section 4.3, also in connection with RESPONSE-3 above.
* * *
Apart from the mentioned points, it would be appropriate to correct some minor errors, such as in the caption of Figure 1 (metre-by-meter). I also suggest mentioning the brand and the type of the devices used for physiological measurements in the relevant section. The resolution of the graphics in the preprint version I downloaded was quite low. I think it would be useful for the readers to check the resolution of the figures used in the text and increase it if possible.

RESPONSE-6: Thank you for these detailed remarks. In the revised manuscript, we would make sure that they are considered.
* * *
**CHANGES MADE**: As announced, we considered all these remarks and corrected/added accordingly. Please note that the devices were self-built, and therefore a brand/model type cannot be given. However, we mention this and direct readers to the relevant reference in the revised MS (Section 3.1).

---

## Author Response (AR2)

Dear Editors,

Thank you very much for your positive response and for suggesting final modifications. We have tried to incorporate as many as possible, except those where we thought the change would not enhance the clarity of the manuscript. See the short explanations below.

We are pleased that our study is now in the position for publication. Thanks again for your help.

Kind regards,

Thomas Mölg, Jan Christoph Schubert, and co-authors

- - - - - - - - - - - - - - - - - - - - - - - - - -

Data availability statement … /// *The website is currently the best place to access the data. We extended the statement to make things clearer*.

Line 41: Please provide translation for the German term. /// *The translation is already in the same sentence. We slightly changed the sentence to express this.*

Line 61: High school students (please insert student age range … /// *Information is on the next page.*

Line 85: Please give 2-3 examples of tree growth variables. /// *This is detailed in Sect. 3.1; we moved the reference to Sect. 3.1 at the end of the sentence to clarify it.*

Line 208: What do you mean by 'talking trees'? // *It was explained before in lines 168-170.*

Line 253: Where are these examples? Consider including them as supplemental materials. /// *These examples changed depending on the time of the workshop. We modified the sentence to clarify the use of real-time data.*

All other suggestions were considered and the respective text was clarified.